# Pharmacogenomic Analysis Reveals *CCNA2* as a Predictive Biomarker of Sensitivity to Polo-Like Kinase I Inhibitor in Gastric Cancer

**DOI:** 10.3390/cancers12061418

**Published:** 2020-05-30

**Authors:** Yunji Lee, Chae Eun Lee, Sejin Oh, Hakhyun Kim, Jooyoung Lee, Sang Bum Kim, Hyun Seok Kim

**Affiliations:** 1Severance Biomedical Research Institute, Yonsei University College of Medicine, Seoul 03722, Korea; dldbswl1110@gmail.com (Y.L.); osj118@yuhs.ac (S.O.); gkrgus951001@gmail.com (H.K.); ljyoung0510@gmail.com (J.L.); 2Brain Korea 21 Plus Project for Medical Science, Yonsei University College of Medicine, Seoul 03722, Korea; 3Department of Medicine, Yonsei University College of Medicine, Seoul 03722, Korea; lce0926@naver.com

**Keywords:** polo-like kinase 1, CCNA2, BI-2536, gastric cancer, KRAS

## Abstract

Despite recent innovations and advances in early diagnosis, the prognosis of advanced gastric cancer remains poor due to a limited number of available therapeutics. Here, we employed pharmacogenomic analysis of 37 gastric cancer cell lines and 1345 small-molecule pharmacological compounds to investigate biomarkers predictive of cytotoxicity among gastric cancer cells to the tested drugs. We discovered that expression of *CCNA2*, encoding cyclin A2, was commonly associated with responses to polo-like kinase 1 (PLK1) inhibitors (BI-2536 and volasertib). We also found that elevated *CCNA2* expression is required to confer sensitivity to PLK1 inhibitors through increased mitotic catastrophe and apoptosis. Further, we demonstrated that *CCNA2* expression is elevated in *KRAS* mutant gastric cancer cell lines and primary tumors, resulting in an increased sensitivity to PLK1 inhibitors. Our study suggests that *CCNA2* is a novel biomarker predictive of sensitivity to PLK1 inhibitors for the treatment of advanced gastric cancer, particularly cases carrying *KRAS* mutation.

## 1. Introduction

Gastric cancer is one of the most common malignant tumors of the gastrointestinal track [1]. Due to complex molecular mechanisms and clinical heterogeneity, clinical outcomes for patients with advanced gastric cancer remain poor, with a 5-year survival of 5–20% and a median overall survival (OS) of 10 months [2].To date, only two targeted therapies, treatment with trastuzumab (HER2 inhibitor) or ramucirumab (VEGFR2 inhibitor), have been approved for the treatment of advanced gastric cancer in patients carrying relevant biomarkers and development of more targeted therapeutic strategies for gastric cancer is needed.

Polo-like kinase 1 (PLK1), a mitotic serine/threonine protein kinase, regulates various cellular events throughout the cell cycle and has been shown to potentially be a new target in cancer treatment [3]. Increased expression of PLK1 has been observed in several types of malignant tumors and has been shown to be correlated with lower survival rates among solid tumor patients [4,5]. Meanwhile, several PLK1 kinase inhibitors have been developed as anticancer drugs and are currently being evaluated in clinical trials [6]: BI-2536, a dihydropteridinone compound and potent ATP-competitive PLK1 inhibitor [7], was found to inhibit cell proliferation in several human cancer cells, including breast, colon, lung, pancreas and prostate cancer [8]. Building on these results, BI-2536 became the first selective PLK1 inhibitor investigated in clinical trials of patients with solid tumors [9] and exhibited an acceptable safety profile in phase I clinical trials. In phase II study, however, BI-2536 showed relatively poor clinical efficacy, with only 4.2% of patients achieving a partial response in treatment of stage IIIB/IV non-small cell lung cancer [10]. Similar clinical data for BI-2536 were observed in another study of advanced solid tumors [11]. In light of these reports, further clinical studies of BI-2536 as a monotherapy have garnered little interest [10,11]. However, identifying a patient selection biomarker may help to overcome the inefficiency associated with BI-2536 monotherapy and assist with identifying patients who may better respond to treatment with PLK1 inhibitors. Indeed, research has shown that *KRAS* mutant cancer cells are highly sensitive to PLK1 inhibition [12], wherein cancer cells carrying the oncogenic mutation *KRAS* were sensitive to PLK1 depletion by shRNA or to treatment with PLK1 inhibitors. However, detailed mechanisms of the actions of PLK1 inhibitors on *KRAS* mutant cancers are largely unknown.

While several drugs targeting *KRAS G12C* mutant cancer sare under clinical trials [13], the *KRASG12C* mutation is very rare in gastric cancer: only 3.6% of *KRAS* mutant gastric cancer patients have the mutation according to combined cohort datasets in the cBioPortal (http://www.cbioportal.org). Therefore, development of alternative therapies will be significant for treatment of *KRAS* mutant gastric cancers.

In this study, we reviewed toxicity screens of 1345 FDA-approved, small-molecule pharmacological compounds and investigational anticancer compounds against a panel of 37 gastric cancer cell lines. Using elastic net regularization, we generated statistical models that predicted the sensitivity of gastric cancer cells to each of the tested drugs based on mRNA expression features, which allowed us to identify distinct drug–biomarker relationships. By focusing on an observed relationship between PLK1 inhibitors and *CCNA2*, we discovered that oncogenic *KRAS* mutation drives *CCNA2* upregulation and consequent mitotic catastrophe and apoptosis in the presence of PLK1 inhibitors.

## 2. Results

### 2.1. Pharmacogenomic Analysis Highlights Novel Drug–Biomarker Relationships Among Gastric Cancer Cells

We previously screened seven gastric cancer cell lines against 1345 pharmaceutical compounds and selected 63 compounds that induced a greater than 50% decrease in cell viability in at least four of the seven cell lines after 72 h of exposure [14]. In this study, we expanded this to 37 gastric cancer cell lines and to 75 compounds targeting cell proliferation, survival and signal transduction pathways (Figure 1a,b). Cell line-specific responses to each of the 75 drugs were calculated by estimating the mean area under survival curves in duplicate (Figure 1c and Appendix A). Using elastic net regularization, we generated statistical models that predicted the sensitivity of gastric cancer cells to each of the tested drugs according to mRNA-based gene expression features. In result, we found 23 biomarkers that predicted sensitivity among gastric cancer cells to nine drugs under bootstrapping (random sampling of cell lines with replacement) and a frequency threshold of 75% (Figure 1d and Appendix A). Intriguingly, *CCNA2*, encoding cyclin A2, which regulates cell cycle progression during the S phase and in G2/M transition, was commonly associated with responses to PLK1 inhibitors BI-2536 and volasertib (BI-6727) (Figure 1d). The concordant associations with *CCNA2* expression (i.e., elevated *CCNA2* predicts hypersensitivity) with two structurally distinct PLK1 inhibitors, but not with other drugs, were suggestive a biologically meaningful relationship. Therefore, we decided to further investigate whether *CCNA2* may be a functional of differential responses to PLK1 inhibitors in gastric cancer.

### 2.2. CCNA2 Upregulation is Causally Linked to BI-2536 Induced Cytotoxicity in Gastric Cancer Cells

First, we sought to validate differential expression of cyclin A2 protein in gastric cancer cell lines selected from both sides of the drug response profile for PLK1 inhibitors. Compared to resistant cells, gastric cancer cells sensitive to PLK1 inhibitors showed increased expression of cyclin A2 (Figure 2a). MKN28 (sensitive) and SNU719 (resistant) cells were further evaluated in regards to multi-point dose responses to BI-2536. As expected, MKN28 cells exhibited greater sensitivity to BI-2536 than *SNU719 cells* (Figure 2b). In MKN28 and other sensitive cancer cell lines (AGS and SNU601), but not in SNU719 cells, BI-2536 elicited PARP1 cleavage, JNK phosphorylation and caspase-3 cleavage, all of which are indicative of apoptosis induction (Figure 2c and Appendix A). To determine if elevated CCNA2 is required to confer sensitivity to BI-2536 in gastric cancer cell lines, CCNA2 was transiently overexpressed in SNU719 cells and knocked down in MKN28 cells (Figure 2d). MKN28 cells transfected with CCNA2 siRNAs gained resistance to BI-2536, compared to cells transfected with negative control siRNA (siNC) (Figure 2e). Meanwhile, however, SNU719 cells transfected with CCNA2 cDNA exhibited greater sensitivity to BI-2536 than control cells (Figure 2f). To confirm that BI-2536 sensitivity indeed acts in relation to CCNA2 expression, we stably knocked down CCNA2 in MKN28 cells using viral transduction of shRNAs and tested the resultant cell viability. Therein, MKN28 isogenic cells, in which CCNA2 was knocked down by five shRNA clones, showed decreased cyclin A2 expression (Figure 2g) and increased viability against BI-2536 (Figure 2h). Similarly, we also observed the shCCNA2-mediated reversal of cytotoxicity to BI-2536 in AGS and SNU601 (Appendix A). Taken together, we deemed that elevated CCNA2 expression is required to confer sensitivity to PLK1inhibitors in gastric cancer cell lines.

### 2.3. CCNA2 is Required for BI-2536-Induced Mitotic Catastrophe and Apoptosis

Cyclin A2 regulates cell cycle progression by promoting S phase entry upon forming a complex with CDK2, as well as by facilitating mitosis through cooperation with the cyclin B1-CDK1 complex [15]. In contrast, PLK1 primarily functions in the M phase of the cell cycle [16,17,18,19] and inhibition of PLK1 causes cell cycle arrest at the G2/M-phase, followed by mitotic catastrophe, a type of apoptosis that occurs during mitosis, in cells with higher mitotic index [20,21]. Therefore, we hypothesized that aberrant upregulation of cyclin A2 in gastric cancer cells may elicit synthetic lethal vulnerability to PLK1 inhibition through failed cell cycle progression, particularly at the M phase. To test this, we performed immunocytochemistry using phospho-histone H3 antibody to detect mitotic cells and found that cyclin A2-knockdown cells show lower mitotic index values than control cells (Figure 3a). To investigate whether elevated cyclin A2 induces sensitivity to BI-2536 due to impaired mitotic progression, we assessed changes in cell numbers in each phase of the cell cycle at 24, 48 and 72 h post treatment with BI-2536 and with or without *CCNA2* knockdown in MKN28 cells. Control cells showed marked cell cycle arrest at the G2/M-phase at 24 h posttreatment with BI-2536; however, shCCNA2 cells slipped over from the G2/M-arrest and showed accumulation of polyploidy at 48 h and 72 h post BI-2536-treatment in a dose-dependent manner (Figure 3b), indicating that cancer cells characterized by high expression of cyclin A2 undergo less mitotic slippage and more apoptosis in response to BI-2536 treatment than cyclin A2 knockdown cells. We confirmed this through flow cytometricanalysis and subsequent immunoblot analysis of control MKN28 cells, which showed early apoptotic cell populations within 24 h of BI-2536 treatment (Appendix A) and accumulation of apoptotic marker proteins (e.g., 89-kDa cleaved PARP1, JNK phosphorylation and cleaved caspase-3) upon exposure to BI-2536 (Figure 3c). shRNA-mediated knockdown of cyclin A2 significantly reduced apoptotic marker proteins (Figure 3c), cell-fractions under mitotic catastrophe (Figure 3d) and dead cell populations (Figure 3e) induced by BI-2536.These observations indicated that PLK1 inhibition in the context of elevated *CCNA2* leads to mitotic catastrophe and apoptosis rather than to cell survival through mitotic slippage. Meanwhile, research has indicated a potential direct regulatory mechanism for cyclin A2 on PLK1 activation and phosphorylation [22]. To investigate if cyclin A2 functions through PLK1 activity, we developedphosphomimetic mutant PLK1 (T210D) and non-phosphorylatable mutant PLK1 (T210A) proteins (Appendix A) and observed their effects on BI-2536 sensitivity. Interestingly, overexpression of neither of these mutant PLK1 proteins nor wild-type PLK1 altered sensitivity to BI-2536 (Appendix A), indicating that *CCNA2*-induced BI-2536 sensitivity is independent of PLK1 and phospho-PLK1 levels.

### 2.4. KRAS Driven Upregulation of CCNA2 Confers Sensitivity to PLK1 Inhibitors Among KRAS Mutant Cancers

While it was reported that *KRAS* mutant cancer cells are highly sensitive to PLK1 inhibitors [12], detailed mechanisms underlying the sensitivity are largely unknown. Here, we hypothesized that oncogenic *KRAS* would drive aberrant upregulation of *CCNA2*. To test this, we compared *CCNA2* expression levels between wild-type and mutant *KRAS* or pan-*RAS* (*KRAS*, *HRAS* and *NRAS*) tumor samples in The Cancer Genome Atlas (TCGA) cohort. *CCNA2* expression was significantly higher in pan-*RAS* and *KRAS* mutant tumors than wild-type controls (*p* =1.01 × 10^−31^ and 4.27 × 10^−20^ by Wilcoxon test, respectively). *KRAS* mutant gastric tumor samples (TCGA-STAD) also showed significantly higher expression of *CCNA2* (*p* = 4.76 × 10^−4^ by Wilcoxon test) than wild-type gastric tumors (Figure 4a). The 37 gastric cancer cell lines, which includedeight *KRAS* mutant cell lines (AGS, SNU601, SK4, SNU1, NCC24, SNU668, YCC2 and NCC59) in this study also showed similar results in that *KRAS* mutant cell lines had higher sensitivity to BI-2536 and volasertib (Figure 4b). To test whether *KRAS* affects *CCNA2* expression or vice versa, *KRAS* mutant gastric cancer cell lines were transfected with siRNAs targeting *KRAS* or *CCNA2*. Therein, the *KRAS* mutant gastric cancer cell lines (AGS, SK4 and SNU601) showed decreased *CCNA2* expression after depletion of mutant *KRAS*, whereas *CCNA2* knockdown did not affect expression of *KRAS* (Figure 4c). In addition, while depletion of mutant *KRAS* reversed cytotoxicity to BI-2536 and volasertib (Figure 4d), co-expression of *CCNA2* in this context was sufficient to reintroduce sensitivity (Figure 4e), indicating that sensitivity to PLK1 inhibitors in *KRAS* mutant gastric cancer cells is mediated by *CCNA2* upregulation.

Taken together, these data demonstrated that oncogenic *KRAS*-driven *CCNA2* upregulation confers hypersensitivity to PLK1 inhibition through mitotic catastrophe and apoptosis (Figure 4f).

## 3. Discussion

The cell cycle is tightly regulated by cyclins, cyclin-dependent kinases (CDKs) and various checkpoint kinases [23]. Deregulation of the cell cycle is a hallmark of tumorigenesis [24]. Given its importance in tumorigenesis, several cell cycle inhibitors have emerged as potential therapeutic drugs for the treatment of cancers, both as single and combination therapies with traditional cytotoxic or molecular targeting agents. Currently, the most promising cell cycle inhibitors in anticancer therapeutics are orally bioavailable CDK4/6 inhibitors, which have received regulatory approval in combination with hormonal therapy for treatment of patients with metastatic hormone receptor (HR)-positive, Her2-negative breast cancer [25,26,27,28,29]. Many other compounds designed to interrupt cell cycle progression or checkpoint control have problems in demonstrating sufficient antitumor efficacy. Thus, to facilitate clinical development of this target class, proper patient selection biomarkers are urgently needed.

PLK1 is a serine/threonine kinase that regulates various cellular events during cell cycle progression, including centrosome maturation, DNA checkpoint activation, mitotic entry, spindle assembly and cytokinesis [3]. Because hyper-activation of PLK1 causes overriding checkpoints, which leads to immature cell division, several PLK1 kinase inhibitors have been developed as anticancer drugs and are currently being evaluated in clinical trials [6]. Several challenges limiting the success of these drugs are as follows: First, most PLK1 inhibitors exhibit a narrow therapeutic window, with their therapeutic effects often coupled with normal toxicity. Second, responses to PLK1 inhibitors are inconsistent. Last is the emergence of drug resistance [6]. Discovery of patient selection biomarkers can help overcome this challenge by identifying correct tumor types and by facilitating patient stratification to increase response rates to these drugs. Indeed, some studies have reported that *TP53* mutation is a biomarker of sensitivity to PLK1 inhibitors [30,31]. This suggests a compensatory mechanism mediated by p53 that rescues cancer cells from mitotic arrest and subsequent apoptosis caused by PLK1 inhibition. Another potential biomarker predicting sensitivity to PLK1 inhibitors is the oncogenic *KRAS* mutation. Research has been shown that cancer cells with *KRAS* mutation are more sensitive to PLK1 inhibitors than *KRAS* wild-type cancers [32], suggesting that *KRAS* mutation induces mitotic stress in tumor cells and may underlie tumor sensitivity to anti-mitotic agents.

Cyclin A2 belongs to the highly conserved cyclin family and is expressed in almost all tissues in the human body. It plays critical roles in control of the cell cycle at G1/S and in G2/M transition. Data from the Human Protein Atlas show that *CCNA2* is overexpressed in dozens of cancer types, suggesting a potential role in tumorigenesis. In this study, we demonstrated that, compared to cells with basal *CCNA2* expression, cancer cells highly expressing *CCNA2* are more addicted to PLK1 activity and show increased mitotic index values, leading to G2/M arrest and mitotic catastrophe, followed by apoptosis, in response to PLK1 inhibitors. We observed neither PLK1 nor phospho-PLK1 affects sensitivity to PLK1 inhibitor, suggesting that the increased sensitivity of *CCNA2*-elevated cancer cells to PLK1 inhibition is not due to direct perturbation of the cyclin A2-PLK1 axis, but rather, likely due to a different mechanism that needs to be further elucidated. One possibility may be that increased cyclin A2 facilitates G2/M transition by activation of the cyclin B1/CDK1 complex [33], resulting in increased mitotic cell populations that more greatly rely on proper spindle assembly checkpoint, wherein PLK1 kinase activity plays an essential role [34].

We also discovered a causal relationship between oncogenic *KRAS* mutation and *CCNA2* upregulation. Our data suggest that aberrations in *CCNA2* expression are a consequence of oncogenic *KRAS* mutation potentially contributing to cell cycle progression, while at the same time, conferring dependence on PLK1 function for productive mitotic progression. Although it is beyond the scope of this manuscript, several mechanistic hypotheses may explain the connections between KRAS mutation, cyclin A2 elevation and PLK1 inhibitor sensitivity: Potentially, upon glutamine (Gln) deprivation, KRAS-driven cancer cells bypass a late G1 Gln-dependent cell cycle checkpoint and enter S-phase, followed by cell cycle arrest due to insufficient nucleotide biosynthesis [35]. Meanwhile, cyclin A2 forms a complex with CDK2 at the S phase of the cell cycle to initiate and progress DNA synthesis. Thus, elevated cyclin A2 in KRAS mutant cancers may reflect anadaptation mechanism from this cell cycle stress at the S phase. Alternatively, as cyclin A2 directly phosphorylates and activates protein kinase B, also known as Akt [36,37], elevated cyclin A2 may contribute to Akt-driven tumorigenesis as well. Either way, as a consequence of *CCNA2* upregulation, cancer cells mayface an unavoidable dependence on PLK1 to prevent mitotic catastrophe by disrupted spindle assembly. Thus, we considerthis relationship as a “synthetic lethality.”Accordingly, synthetic lethal approaches targeting cell cycle progression with PLK1 inhibitors may prove to be effective in treating tumors characterized by increased *CCNA2* expression.

## 4. Materials and Methods

### 4.1. Pharmacological Characterization

This study analyzed data for 37 gastric cancer cell lines treated with 75 small-molecule compounds selected from libraries of FDA-approved small-molecule pharmacological compounds (#L1300, Selleckchem, Houston, TX, USA) and investigational anti-cancer compounds (#L2000, Selleckchem). The pharmacological profiles of 29 of the 37 cell lines have previously been reported [14]. For cell-based drug assay, 5000 cells were seeded onto individual wells of 96-well plates. After 24 h of incubation, half-log 12-serial dilutions of pharmacological compounds in DMSO were added using a BiomekFXp liquid handler (Beckman Coulter, Brea, CA, USA), resulting in final concentrations of 50 μM to 0.5 nM. The cells were then incubated for 72 h and cell viability was measured with CellTiter-Glo assay kits (Promega, Madison, WI, USA). In each cell line, DMSO (0.5%) controls were used for normalization. Finally, we calculated area under the viability curve (AUC) values from 12-point dose–response curves for each pharmacological compound. All gastric cancer cell lines, except SK4 and the Yonsei Cancer Center (YCC)-series cell lines, were purchased from the Korea Cell Line Bank. SK4 cells were kindly provided from Dr. Julie Izzo (MD Anderson Cancer Center, Houston, TX, USA). YCC-series cell lines were obtained from the Song–Dang Institute for Cancer Research (Yonsei University College of Medicine, Seoul, Korea). The cell lines were maintained in RPMI-1640 medium supplemented with 10% fetal bovine serum (Gibco/Thermo Fisher Scientific, Waltham, MA, USA) and 1% penicillin–streptomycin (Invitrogen, Carlsbad, CA, USA) in mycoplasma-free condition. All gastric cancer cell lines have been authenticated using STR profiling within the last three years.

### 4.2. RNA Sequencing

RNA sequencing (RNA-seq) data for 29 of the 37 gastric cancer cell lines were previously reported [14]. Total RNA from the eight remaining gastric cancer cell lines were extracted with a RNeasy Plus Mini Kit (Qiagen, Hilden, Germany). RNA-seq libraries were then generated with a TruSeq RNA Sample Prep kit v2 (Illumina, San Diego, CA, USA) and sequencing with the HiSeq 2500 platform. The TopHat-Cufflinks pipeline was used to align reads to the reference genome and to calculate normalized values in FPKM (fragments per kilobase of exon per million fragments mapped).

### 4.3. Pharmacogenomic Analysis

We established elastic net models using a previously described method with some modifications [38]. Briefly, transcriptome and drug response data (*n* = 75) for the 37 gastric cancer cell lines were used to build the model. To do so, gene expression values were converted into Z-scores. Optimal values of α and λ were determined by 10-fold cross validation from 100 iterations. Bootstrapping (200×) was applied to estimate average weights (β) and selection frequency of features by the model. For drug marker selection, we chose features occurring at a frequency > 75%. Next, we applied different weight cutoffs to individual drugs because their weight spectrumsvaried greatly, which made it difficult to apply a single weight cutoff. The feature selection process was conducted using the GlmnetR package (version 2.0–8) and R (version 3.3.3).

### 4.4. siRNA Transfection

In each well of 96-well plates, 30 µL of 333 nM siRNA solution was mixed with 10 µL of 2% RNAiMAX (Invitrogen) solution and incubated for 15 min. Subsequently, 7000 cells in 100 µL of growth medium were added to the mixture. Culture medium was replaced 24 h post transfection. siRNA oligonucleotides were custom synthesized (Genolution, Seoul, Korea) with the sequences: 5′-GAUAUACCCUGGAAAGUCUUU-3′ (siCCNA2-1), 5′-GGAUGGUAGUUUUGAGUCAUU-3′ (siCCNA2-2), 5′-CUAUGGACAUGUCAAUUGUUU-3′ (siCCNA2-3), 5′-CGAAUAUGAUCCAACAAUAUU-3′ (siKRAS-1), 5′-GACAAAGUGUGUAAUUAUGUU-3′ (siKRAS-2), 5′-GCAUGGGACAUUUGUGAUUUU-3′ (siNC).

### 4.5. cDNA Transfection

The Myc-DDK-tagged human CCNA2 plasmid (#RC211148L1) was purchased from OriGene (Rockville, MD, USA). Two million cells were grown in 60-mm dishes for 24 h. On the day of transfection, 5 µg of plasmids and 15 µL of Lipofectamine 2000 (Invitrogen, Carlsbad, CA, USA) were diluted in 500 µL of Opti-MEM and added onto the culture dishes. Twenty-four hourspost transfection, cell lines transfected with either target plasmid or empty vector were trypsinized and re-plated into 96-well plates (8000 cells per well) for drug toxicity assay.

### 4.6. Immunoblot Analysis

Total cell extracts were prepared by dissolving cells with 1×Laemmli SDS reducing buffer (50 mM Tris-HCL [pH 6.8], 2% SDS and 10% glycerol) and boiled for denaturation. Equal amounts of protein sample were separated on 4-15% Mini-PROTEAN TGX^TM^ Precast Gel (Bio-Rad, Hercules, CA, USA). Anti-Cyclin A2 (#4656S, Cell Signaling, Danvers, MA, USA), anti-β-actin (#sc-47778, Santa Cruz Biotechnology, Dallas, TX, USA), anti-KRAS (#sc-30, Santa Cruz Biotechnology), anti-cleaved PARP (#9541S, Cell Signaling Technology), anti-phospho-JNK (#4668S, Cell Signaling Technology), anti-cleaved Caspase-3 (#9661S, Cell Signaling Technology), anti-PLK1 (#4513S, Cell Signaling Technology), anti-FLAG (DYKDDDDK)(#2368S, Cell Signaling Technology) and anti-GAPDH (#60004-1-Ig, Proteintech Group, Rosemont, IL) antibodies were used as primary antibodies. Peroxidase-AffiniPure Goat Anti-Rabbit IgG (#111-035-144) and Anti-Mouse IgG (#115–035–003, Jackson ImmunoResearch, West Grove, RA, USA) were used as secondary antibodies. Antibody binding was visualized by SuperSignal West Pico/FemtoChemiluminescent Substrate (Thermo Fisher Scientific, Waltham, MA, USA) and X-ray films (AGFA-Gevaert, Mortsel, Belgium).

### 4.7. The Cancer Genome Atlas Analysis

Gene expression and mutation data for 16 cancer types were downloaded from the Genomic Data Commons data portal (https://portal.gdc.cancer.gov). Gene expression data in FPKM were transformed to log2 scale values and quantile normalization was performed to remove technical biases. Samples bearing any mutations in *KRAS*, *HRAS* or *NRAS* were considered as RAS mutation samples. A two-sided Wilcoxon rank-sum test was used to test the difference in *CCNA2* expression levels between RNA-mutant and wild-type samples.

### 4.8. Flow CytometricAnalysis

For cell cycle analysis using flow cytometry (FACSVerse^TM^, BD Biosciences, Franklin Lakes, NJ), MKN28 cells were plated at a density of 1 × 10^3^ cells in 60mm culture dishes and then treated with BI-2536 (200 and 500 nM) for 72 h. Cells were then harvested and fixed in ice-cold 70% ethanol overnight at −20°C. Afterwards, cells were centrifuged at 300× g for 5 min, re-suspended in PBS containing 10 µg/mL of propidium iodide (PI) (P4170, Sigma-Aldrich, St. Louis, MO, USA), 100 µg/mL RNase A and 0.1% (*v/v*) Triton X-100 and incubated at 37 °C for 10 min. For cell apoptosis assay of MKN28 cells treated with BI-2536 (200 and 500 nM), we used the Annexin V-FITC Apoptosis Kit (#640914, BioLegend, San Diego, CA, USA) according to the manufacturer’s protocol. Briefly, a total of 1 × 10^3^ cells was seeded in 60mm culture dishes for 24 h and then treated with BI-2536 for 72 h. Adherent and floating cells were then harvested, stained with Annexin V and PI for 15 min at room temperature and subjected to flow cytometry. The data were analyzed with FlowJo software.

### 4.9. Immunofluorescence Analysis

Cells were washed with ice-cold PBS and fixed with 3.7% paraformaldehyde (PFA) for 10 min, permeabilized with 0.5% Triton X-100 in PBS (PBS-T) for 5 min and incubated with blocking solution (0.1% BSA + 10% goat serum in 0.1% PBS-T) for 30 min. Cells were then incubated with primary antibodies diluted in 0.1% PBS-T for an hour. After incubation with secondary antibodies labeled with Alexa Fluor-488 or Alexa Fluor-568 (Invitrogen), coverslips were mounted on slide glasses using Prolong Gold Antifade mounting solution (Thermo Fisher Scientific) and the slides were allowed to dry at room temperature.

## 5. Conclusions

We performed pharmacogenomics analysis using a gastric cancer cell line panel and discovered a causal linkage cascade of oncogenic *KRAS* mutation, aberrant *CCNA2* upregulation and hypersensitivity to PLK1 inhibitors. Our findings hold translational implications for the treatment of gastric cancer patients with aberrant upregulation of *CCNA2* via synthetic lethal approaches targeting cell cycle progression.

## Figures and Tables

**Figure 1 cancers-12-01418-f001:**
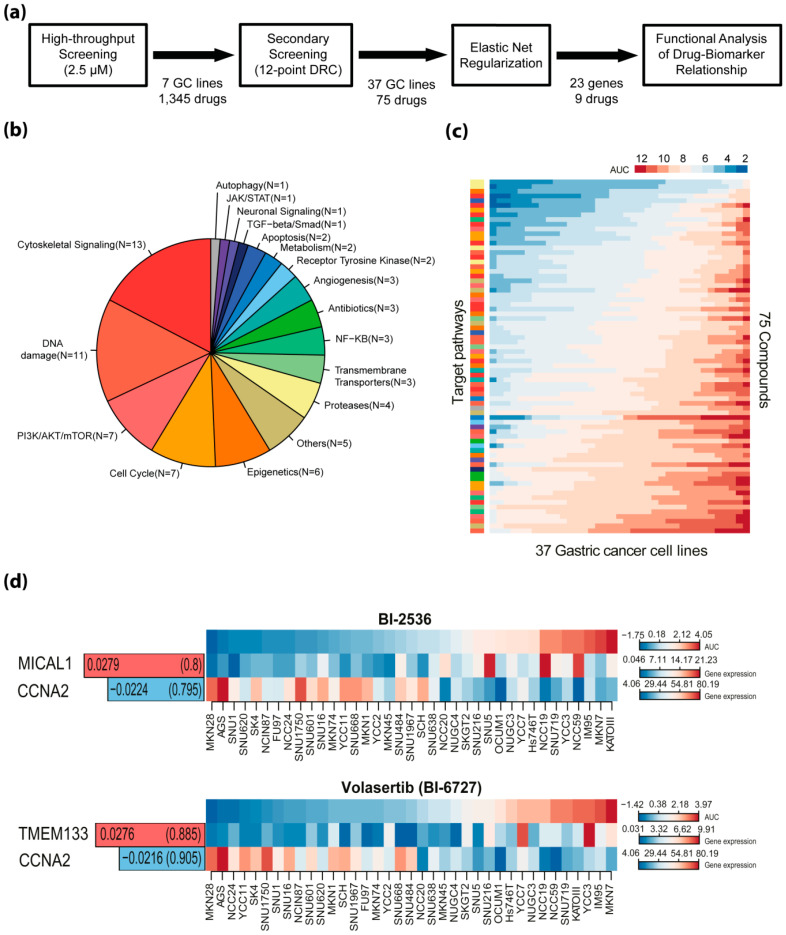
Pharmacogenomic analysis identifies biomarker–drug response relationships. (**a**) Flowchart of overall screening strategy; (**b**) classification of the 75 compounds according to their target pathways; (**c**) sensitivities (area under the viability curve (AUC)) of the 37 gastric cancer cell lines to 75 compounds are ordered by row. Rank-ordered original AUC values are indicated as a heat map. Heat mapsare colored on a blue (sensitive) to white to red (resistant) gradient scale of original AUC values. Target pathways for each compound are annotated by the same color code as in b; (**d**) representative biomarker and drug response relationships by elastic net regularization method. The average weights of features are displayed in bar plots and their frequencies are shown in parenthesis.Bar plots on the left are colored in red when the expression level of a biomarker is positively correlated with the resistance of the given drugs and colored in blue when negatively correlated. Heat mapsaredepicted on a blue–white–red gradient scale of median-centered AUC values and expression levels (FPKM) of genes, respectively.

**Figure 2 cancers-12-01418-f002:**
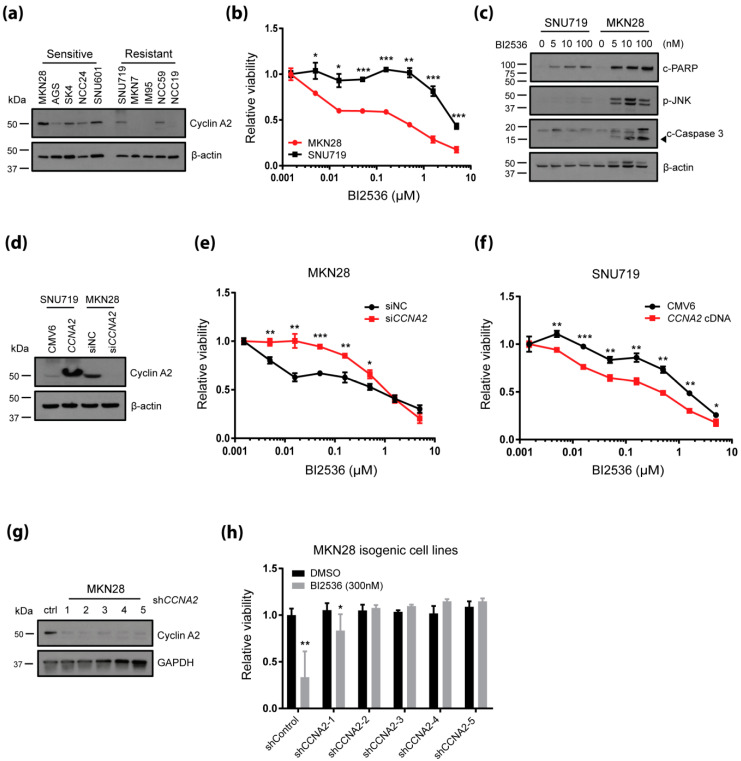
Elevated *CCNA2* is required to confer sensitivity to BI-2536. (**a**) Expression levels of endogenous cyclin A2 were assessed by immunoblotting whole cell lysates from the indicated “resistant” and “sensitive” gastric cancer cell lines; (**b**) dose–response curves of cell viability for the indicated gastric cancer cell lines after 72 h of exposure to BI-2536; (**c**) Induction of apoptotic markers were assessed by immunoblotting. Whole cell lysates were prepared post 72 h of BI-2536 or vehicle (DMSO) treatment with indicated concentrations; (**d**) ectopic expression of *CCNA2* in SNU719 cells and knockdown of *CCNA2* in MKN28 cells were demonstrated by immunoblotting; (**e**) dose–response curves of MKN28 cells expressing non-silencing siRNA (siNC) or siRNA against *CCNA2* (siCCNA2); (**f**) Dose–response curves of SNU719 cells transfected with empty pCMV6 plasmid or *CCNA2* cDNA plasmid; (**g**) immunoblot shows depletion of cyclin A2 in MKN28 cells expressing shRNA clones against *CCNA2*; (**h**) relative viability of MKN28 cells at 72 h post BI-2536 (300 nM) treatment; (b,e,f) * *p* <0.05, ** *p* < 0.01, *** *p* <0.001; one-way ANOVA; (h) * *p* <0.05, ** *p*< 0.01, *** *p* <0.001; Wilcoxon rank-sum test.

**Figure 3 cancers-12-01418-f003:**
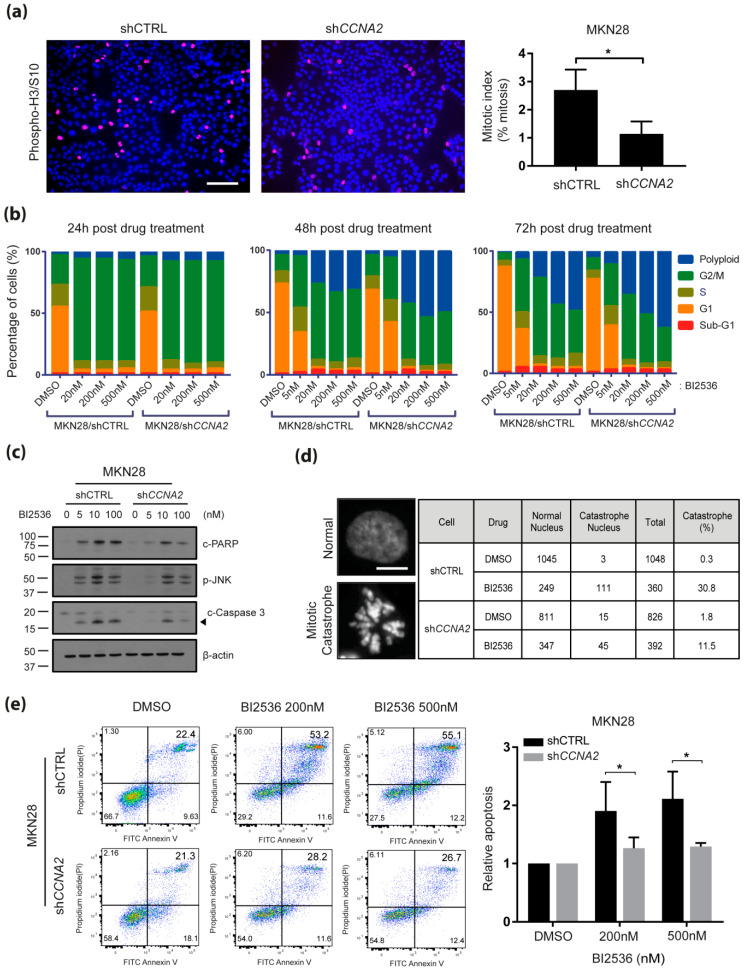
Elevated *CCNA2* is required for BI-2536-induced mitotic catastrophe and apoptosis. (**a**) Mitotic cells were visualized by immunostaining using anti-phospho-Histone H3 antibody (red) in MKN28 cell lines expressing shCCNA2 or shCTRL (left). DAPI was counterstained to detect nuclei (blue). Scale bar: 200 µm. Mitotic index (right), calculated by dividing the number of mitotic cells by the total number of cells. * *p* < 0.05; Student’s *t*-test; (**b**) evaluation of cell cycle by propidium iodide (PI) staining and flow cytometer analysis after BI-2536 treatment for the indicated time periods. Percentages of cells in each cell cycle are presented in bar plots; (**c**) effects of BI-2536 on the expression of apoptotic markers (e.g., cleaved PARP, phospho-JNK and cleaved caspase-3) 72 h post treatment. The amount of β-actin was measured as an internal control; (**d**) evaluation of mitotic catastrophe by DAPI staining and fluorescence microscopy (Axio Imager M2, ZEISS) at 72 h post BI-2536 (300 nM) treatment (Scale bar, 25µm); (**e**) apoptosis analysis using Annexin V-FITC/PI dual staining and flow cytometer analysis after BI-2536 (200 nM and 500 nM) treatment for 72 h are presented in both scatter plots (left) and bar plots (right). * *p* < 0.05; Wilcoxon rank-sum test.

**Figure 4 cancers-12-01418-f004:**
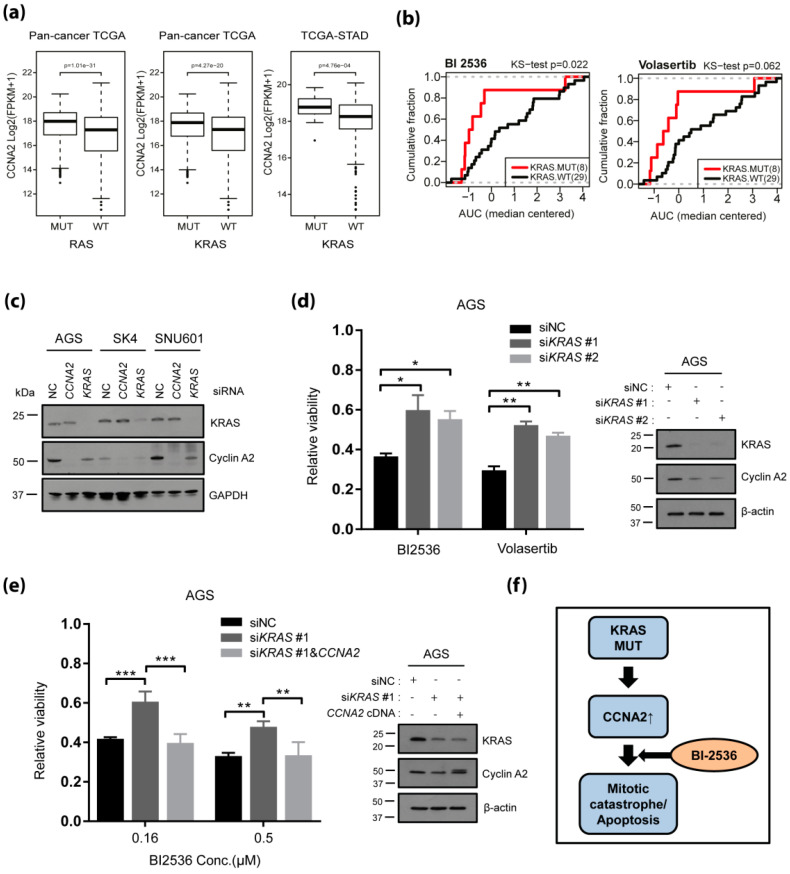
Oncogenic *KRAS* driven *CCNA2* upregulation confers sensitivity of *KRAS* mutant cancer to PLK1 inhibitors.(**a**) Comparison of *CCNA2* expression levels between wild-type (WT) and pan-*RAS* (*KRAS*, *HRAS* and *NRAS*) or mutant *KRAS* tumor samples (pan-cancer and gastric cancer) in the Cancer Genome Atlas (TCGA) cohort; (**b**) cumulative distribution fraction plots of drug response (median-centered AUC) in the 37 gastric cancer cell lines show that *KRAS* mutant cell lines had higher sensitivity to BI-2536 and volasertib. *p* values were calculated by two-sided Kolmogorov–Smirnov tests (KS-test); (**c**) evaluation of cyclin A2 and KRAS expression by immunoblotting post knockdown of *CCNA2* and *KRAS* in *KRAS* mutant gastric cancer cell lines (AGS, SK4 and SNU601). GAPDH was measured as an internal control; (**d**) relative viability of AGS cells expressing siKRASoligos at 72 h post treatment with BI-2536 (0.16μM) and volasertib (0.16μM). Expression changes of KRAS and cyclin A2 by expression of siKRASoligos were observed by immunoblotting. * *p* <0.05, ** *p* <0.01; Student’s *t*-test; (**e**) relative viability of AGS cells expressing siKRAS with or without *CCNA2* cDNA at 72 h post BI-2536 (0.16μM and 0.5μM) treatment. Expression changes of KRAS and cyclin A2 were assessed by immunoblotting. ** *p* <0.01, *** *p* <0.001; two-way ANOVA; (**f**) Hypothetical model of selective toxicity to PLK1 inhibitor in *KRAS* mutant cancer.

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
