# Peer review of "Pharmacogenomic Analysis Reveals CCNA2 as a Predictive Biomarker of Sensitivity to Polo-Like Kinase I Inhibitor in Gastric Cancer"

_cancers, 2020, doi:10.3390/cancers12061418_

Round 1

Reviewer 1 Report

The authors have meticulously addressed all concerns.

Reviewer 2 Report

The authors carefully addressed the raised issues and I do not have further concern regarding the acceptance of this manuscript.

Reviewer 3 Report

All concerns has been addressed, ready for acceptance.

This manuscript is a resubmission of an earlier submission. The following is a list of the peer review reports and author responses from that submission.

Round 1

Reviewer 1 Report

Lee & Lee et al. have prepared a very good and professionally written manuscript. Using high-throughput screening and elastic regression, the authors have identified cyclin A2 as a prognostic biomarker for increased gastric cancer cell line susceptibility to polo-like kinase 1 (Plk1) inhibitors and suggested that these cells undergo death by mitotic catastrophe in
a cyclin A2-dependent manner. In addition, this study has found that levels of cyclin A2 positively correlate with and can be modulated by the levels of mutant KRAS and this could be potentially exploited for anti-mitotic therapy of gastric cancer. Despite the robustness and high quality of the data presented, the conclusion that the mechanism of cell death proceeds through apoptosis following BI-2536 Plk1 inhibitor treatment is not sufficiently supported. Authors are therefore given the choice either to alter their working hypothesis or present new evidence for early apoptotic markers induced by BI-2536 (see below).

Major points:

1) There is a problem in the interpretation of the Annexin V-FITC/PI result in Figure 3C (line 153). Although the flow cytometry analysis, calculated from the Annexin V-FITC positive and PI positive quadrant, refers to "Relative apoptosis", this assay can, in fact, distinguish only between healthy, early apoptotic, and mixed late apoptotic plus necrotic (dead) cell populations (see for example the BioLegend's Technical Data Sheet for catalog #640914). It is therefore formally incorrect to label and interpret the read-out simply as apoptosis. The conclusion that cyclin A2 is indeed required for BI-2536-induced apoptosis following mitotic catastrophe should be strengthened by investigating markers of early apoptosis including repeating the Annexin V-FITC/PI measurement at an earlier time point (at which cells display mitotic catastrophe). Increased percentage of cells found within the Annexin V-FITC positive and PI negative quadrant at this earlier time point would confirm the progression of apoptosis.

Minor points:

1) The caption should read "(12-point DRC)" instead of "(12-points DRC)" and there are extra spaces between "of" and "Drug-Biomarker" in the caption "Functional Analysis of Drug-Biomarker Relationship" in Figure 1A (line 87). Please revise.

2) Please indicate more clearly in the legends to Figures 1C, 1D (line 87), and S1 (lines 338–342 and 448) the meaning of red vs. blue color coding for both cell line drug sensitivity and biomarker association.

3) Please indicate statistical significance using asterisk(s) in Figures 2B and 2H (line 118).

4) Please indicate respective confidence intervals assigned to asterisk symbols in the legends to Figures 2E, 2F (and 2B and 2H; see minor point 3) (line 118), and 3A (line 153) and the statistical test used to derive these values.

5) It is not clear whether regular flow cytometry or fluorescence-activated cell sorting (FACS) technique was used in cell-based assays (lines 147 and 311). Please remove these inconsistencies.

6) Please include a scale bar in at least one image in each of the Figures 3A and 3E (line 153) with length unit definition in the corresponding figure legends.

7) Could the authors comment on in the text on or investigate further the contradiction why the result obtained in Figure 3A (diminished mitotic index) is not recapitulated in Figure 3B (no change in G2/M phase proportion) following cyclin A2 knockdown (line 153)?

8) The propidium iodide (PI) channel is labeled as "PE-A" in the y axes of Figure 3C (line 153). Please either explain the rationale behind using "PE-A" channel in the corresponding figure legend or relabel y axes to avoid confusion between PI and PE.

9) Could the authors please indicate in the corresponding figure legend at what time point was the analysis of apoptotic marker expression in Figure 3D (line 153) performed following BI-2536 treatment?

10) It is stated that 8 out of 37 gastric cancer cell lines used were KRAS mutant (line 176). In addition to 3 of these cell lines specified (AGS, SK4, and SNU601) (line 180), would it be possible to have the identity of all 8 KRAS mutant cell lines indicated either in the text, in a new or existing table (Table S1), or as part of an existing figure?

11) Please define the meaning of "KS-test" in the legend to Figure 4B (line 188).

12) The x axes to Figure 4B (line 188) claim that AUC were median centered. Please indicate whether the same holds true for AUC values generated for Figure 1C (line 87).

13) Please round the BI-2536 concentration used to a reasonable number of digits in the x axis to Figure 4E (line 188).

14) Please replace "Seleckchem" with proper name of the pharmaceutical company Selleckchem (line 250 2x).

15) Could the authors please provide company location (city, state) at the first occurence of the following companies: Selleckchem (line 250), Beckman (line 254), Promega (line 256), Gibco (line 263), Invitrogen (line 263), Qiagen (line 268), Illumina (line 269), Genolution (line 286), Origene (line 292), Bio-Rad (line 300), Cell Signaling (301), Thermo Scientific (303), AGFA (line 303), Sigma-Aldrich (line 316), and BioLegend (line 318)? Please remove the affiliation for Invitrogen "(Carlsbad, CA)" from line 294 since its first occurence is on line 263.

16) Would it please be possible to have all primary and secondary antibodies used in the study and their respective catalog numbers summarized in a table?

17) From the composition of the Laemmli SDS reducing buffer (page 299) is not clear whether immunoblotting was performed under reducing or non-reducing conditions? Please provide the identity of the reducing agent and its concentration, if this reagent was added to the immunoblotted samples.

18) The sentence "The following antibody was used: mouse monoclonal anti-Cyclin A2 antibody (Cell Signaling)" is rather awkward and the wording could be improved (line 300).

19) Could the authors please specify whether adherent only or adherent plus floating (dead) cell fraction was harvested for Annexin V and PI flow cytometry assay in the respective Methods section (lines 317–321)?

20) The manufacturer and its location (city, state) is missing for Prolong Gold Antifade mounting solution (line 328). Please fix.

21) "Identified biomarker and drug response relationships" seems to be duplicated in the same sentence twice (lines 338 and 449).

Reviewer 2 Report

This manuscript suggests CCNA2 as a predictive marker for the sensitivity of PLK1 inhibitors to kill gastric cancer cells. The authors demonstrate that the level of CCNA2 positively correlates with ability of PLK1 inhibitors to mediate cancer cell apoptosis. The findings here would be of interest to the readership of the journal and informative to the field of cancer therapy.  

  1. While the positive correlation is well noted, the statement that “CCNA2 is necessary and sufficient to confer sensitivity to BI-2536 in gastric cancer cells” seems a little far-fetching. Sensitivity may be lower in CCNA2-deficient or null cells, but is unlikely completely lost.
  2. What would be the mechanism(s) by which CCNA2 elevation enhances the sensitivity of PLK1 inhibitors? Have the authors evaluated the PLK1 and phosphorylated PLK1 levels in CCNA2 up- and downregulated gastric cancer cells as well as in presence of PLK1 inhibitors?
  3. The authors hypothesize that cyclin A2 in gastric cancer cells might elicit synthetic lethality vulnerability of PLK1 inhibition. However, it has been reported that cyclin A2 serves as upstream of or directly regulates activation of PLK1. To this end, the authors should provide additional evidence supporting their hypothesis that the observation is due likely to synthetic lethality rather than the perturbation of the cyclin A2-PLK1 axis.

Reviewer 3 Report

Nice work from Dr. Kim and group elaborating the role of cyclin A2 as biomarker in case of sensitivity to PLK1 inhibitor in KRAS mutants. Few things should be addressed before it is ready for acceptance. They are as follows:

  1. It's been shown that cell cycle regulators play a significant role in KRAS mutants. It's been discussed in PMID: 26682255. Authors should add few lines in discussion stating their perspective on how PLK1 will be fitted in cell cycle check-points by referring the mentioned work.
  2. AKT is one of the predictive marker of Cyclin A2 as mentioned in PMID: 24797432. It will be interesting to see author's point of view on this context. Few lines should be added in discussion.
  3. Fig 3A, author should mention the scale of the microscopic pictures in figure legends.  
  4. Few lines should be added in introduction mentioning about KRAS in gastric cancer suggesting that as KRAS inhibitor is not a possible reach at this moment this alternate therapy will be significant for therapeutic purposes.

Reviewer 4 Report

In the manuscript "Pharmacogenomic analysis reveals CCNA2 as a predictive biomarker of sensitivity to polo-like kinase I inhibitor in gastric cancer", the authors attempt to convince the reader that PLKI inhibitors are a viable treatment for patients suffering from gastric tumors that over express CYCLINA2.

This study is well written and easy to understand. The manuscript is also of general interest to the cancer community in general as CYCLINA2 is found to over-expressed in many different types of tumors.

The experimental design appears to be correct.

Concerns:

The staining present in figure 3A is not very good and makes the it hard to see how the quantification was accomplished. Better representive images should be included in the manuscript.
